# Selective Object Rearrangement in Clutter

**Bingjie Tang**
Department of Computer Science
University of Southern California
United States
bingjiet@usc.edu

**Gaurav S. Sukhatme**[*]
Department of Computer Science
University of Southern California
United States
gaurav@usc.edu

**Abstract:** We propose an image-based, learned method for selective tabletop object rearrangement in clutter using a parallel jaw gripper. Our method consists of three stages: graph-based object sequencing (which object to move), feature-based action selection (whether to push or grasp, and at what position and orientation) and a visual correspondence-based placement policy (where to place a grasped object). Experiments show that this decomposition works well in challenging settings requiring the robot to begin with an initially cluttered scene, selecting only the objects that need to be rearranged while discarding others, and dealing with cases where the goal location for an object is already occupied – making it the *first* system to address all these *concurrently* in a purely image-based setting. We also achieve an ∼8% improvement in task success rate over the previously best reported result that handles *both* translation and orientation in less restrictive (uncluttered, non-selective) settings. We demonstrate zero-shot transfer of our system solely trained in simulation to a real robot selectively rearranging up to 15 everyday objects, many unseen during learning, on a crowded tabletop. Videos: https://sites.google.com/view/selective-rearrangement.

**Keywords:** Rearrangement, Robot Manipulation, Task and Motion Planning

## 1 Introduction

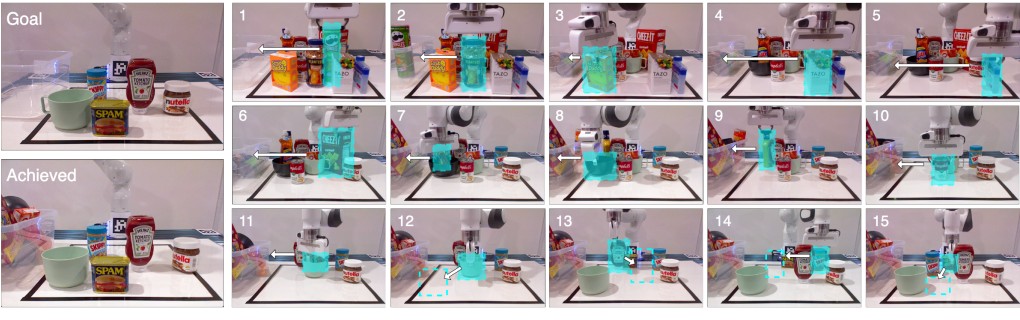

Selected Object ⟸ Moving Direction ⸬ Target Position * All non-target objects moved to side bin. No target position specified.

Figure 1: **15-object selective rearrangement from a cluttered initial state.** Given an initial arrangement of everyday objects and an image specifying the goal arrangement, the robot learns to remove objects that do not need repositioning (1-11) and repositions all other objects accurately (12-15) as specified by the goal image (top left) resulting in the final arrangement (bottom left).

Repositioning objects to a desired configuration is rooted in the activities of daily living [1]. Many skills underlie this capability – extracting useful information from raw perceptual data, performing accurate object manipulation, and optimizing long-term sequential action planning – making object rearrangement an essential challenge for both robotics and embodied AI [2]. Figure 1 illustrates our

---

[*]Gaurav Sukhatme holds concurrent appointments as a Professor at USC and as an Amazon Scholar. This paper describes work performed at USC and is not associated with Amazon.

6th Conference on Robot Learning (CoRL 2022), Auckland, New Zealand.

setting: faced with a tabletop with many everyday objects (**clutter**) the robot is tasked to rearrange a subset of objects (**selectivity**) to a goal configuration, while discarding others in a bin. Another challenge is in situations where the desired locations for some objects are already occupied (**swap**). Object rearrangement has been studied in the context of both task and motion planning and learning. However, existing methods do not *concurrently* address these three challenges. Our system is the first to do so in a purely learned setting where the goal is given by a single RGB-D image.

In contrast to e.g., suction mechanisms, we use a parallel jaw gripper requiring object singulation before grasping. Our method consists of three stages: graph-based **object sequencing** that picks the next object to manipulate by minimizing the Graph Edit Distance (GED) between the current scene graph and the goal scene graph, feature-based **action selection** that maps the RGB-D image to robot actions (pushing or grasping) through a deep Q-learning framework, and a visual correspondence-based **placement policy** that uses the cross-correlation of visual features between the grasped object and the goal specification image to locate object placement. Experiments show the successful rearrangement of 3-7 objects with $>90\%$ completion (error $<2.99$ cm), and 16-20 objects with $>82.33\%$ success (error $<1.64$ cm). We also achieve an $\sim8\%$ improvement in task success rate over the previously best reported result that handles *both* translation and orientation in a less restrictive setting (uncluttered, non-selective). We demonstrate zero-shot transfer to a real robot (Figure 1) selectively rearranging up to 15 everyday objects, many unseen during learning, on a crowded tabletop.

## 2 Related Work

**Task and motion planning**-based systems (TAMP) [3] either have a high-level task planner and a low-level motion planner [4, 5, 6, 7, 8, 9, 10], or use sampling-based algorithms or optimization to solve a single unified formulation of the problem [11, 12]. Some TAMP solutions rely on known object models or a known environment [13, 9, 10], which makes it difficult to deploy them with novel objects or where explicit object pose estimation is difficult to obtain. TAMP approaches that incorporate learning-based vision models, such as [14, 15, 16, 17] can adapt to novel objects/environments while [14] is based on one initial scene image, [15] uses structural constrained predicates for planning, [16] depends on the knowledge of the environment, and [17] assumes round collision radius for all object shapes, making it difficult to scale to adversarial environments (e.g., highly cluttered). The number of possible action sequences increases exponentially with the number of objects and changes in environment observability increase the difficulty of back-tracking and replanning.

**Deep learning**-based systems have relaxed some of these constraints by incorporating learning-based models in perception, planning and actuation. They have been shown to learn general policies to handle varied rearrangement tasks [27, 28, 29, 30, 31, 32], e.g., highly-cluttered, partially-observable environments or deformable objects. Our work is related to learning-based methods for grasping in clutter [18, 19, 20, 33], target object retrieval [21, 22, 23], and rearrangement [24, 25, 10, 26] (Table 1). Most related to our work, Zeng et al. [20] and Tang et al. [33] proposed using deep Q-learning to learn synergies between push and

| Method | Robot Action | Clutter | Selectivity |
|---|---|---|---|
| **Grasping** | | | |
| DexNet [18] | GRASP | ✔ | ✗ |
| GPD [19] | GRASP | ✔ | ✗ |
| VPG [20] | PUSH&GRASP | ✔ | ✗ |
| **Target object retrieval** | | | |
| Mech Search [21] | GRASP | ✔ | ✔ |
| Murali et al. [22] | GRASP | ✔ | ✔ |
| MORE [23] | PUSH&GRASP | ✔ | ✔ |
| **Rearrangement** | | | |
| NeRP [24] | GRASP | ✗ | ✗ |
| IFOR [25] | GRASP | ✗ | ✗ |
| TRLB [10] | SUCTION | ✔ | ✗ |
| ReorientBot [26] | SUCTION | ✔ | ✗ |
| Ours | PUSH&GRASP | ✔ | ✔ |

Table 1: **Related Manipulation Tasks**

grasp to improve grasping accuracy in densely cluttered environment. Inspired by [20] and [33], we adapt collaborative PUSH and GRASP to solve highly cluttered environments for object rearrangement tasks. Different from previous works, we learn action primitives, distinguish objects to rearrange from those to discard, and plan sequential actions simultaneously making this the first work to concurrently solve image-based selective object rearrangement in a cluttered tabletop environment.

## 3 Learning a Selective Rearrangement Policy

We decompose the rearrangement problem into three parts: object sequencing (*which* object to relocate next), action selection (*how* to manipulate it), and object placement (*where* to place a grasped object). We rely on three primitives: pushing objects (PUSH), picking them up (GRASP), and placing

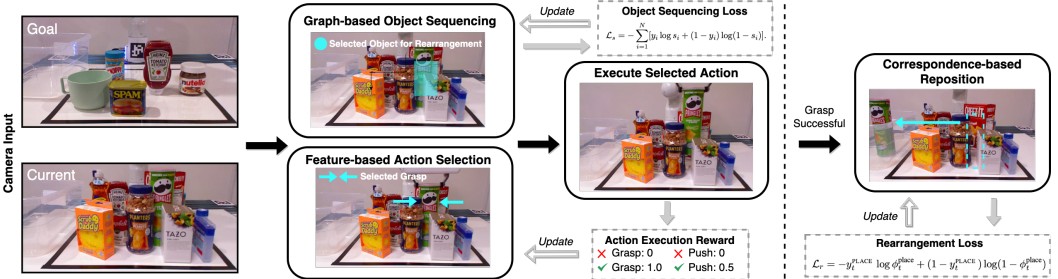

Figure 2: **System overview.** Our system uses RGB-D images as input and builds a scene graph based on the object segmentation given by UOIS-Net-3D [34]. Graph-based object sequencing (subsection 3.3) selects the optimal object for next rearrangement and we mask the Q-value map for GRASP with its segmentation mask. Then the system picks the highest Q-value action candidate from PUSH and GRASP Q-value maps and executes the action Figure 3a. If GRASP is chosen and successfully executed, the system locates the PLACE of the grasped object (Figure 3b).

them at target locations (PLACE). PUSH and GRASP can be initiated by the robot at any time, however PLACE can only be performed if the robot is already holding an object. This suggests a natural decomposition into our three part strategy. When the robot is not holding an object, it must make a decision on which object to manipulate next (object sequencing). After choosing an object, it must decide whether (and how) to push the selected object or whether (and how) to pick it up (action selection). When holding an object, it must decide where to place it (object placement). We model object sequencing as a supervised learning problem on graph transformations (subsection 3.3), action selection as a Partially Observable Markov Decision Process (POMDP) (subsection 3.1), and object placement as a supervised learning problem (subsection 3.2).

## 3.1 Feature-based Action Selection: PUSH or GRASP

The choice of whether to PUSH or GRASP (and at what location and orientation) is Markovian since it is based solely on the current state (object poses). Further, the state is partially observable – we do not assume the robot has direct access to full state information, it needs to be inferred from images. Hence, we formulate the problem of selecting whether to push or pick up an object (and at what location and orientation) as a goal-conditioned POMDP – a tuple $(\mathcal{S}, \mathcal{G}, \mathcal{A}, p, \mathcal{R}, \Omega, \mathcal{O}, \gamma, \rho_0, \rho_g)$ where $\mathcal{S}$ is the state space, $\mathcal{G}$ is the set of goals, $\mathcal{A}$ is the action space, $p(\mathbf{s_{t+1}}|\mathbf{s_t}, \mathbf{a_t})$ is the time-invariant (unknown) dynamics function, $R : S \times A \rightarrow \mathbb{R}$ is the reward function, $\Omega$ is a set of observations, $O$ is a set of conditional observation probabilities, $\gamma \in [0, 1]$ is the discount factor, $\rho_0$ is the initial state distribution, and $\rho_g$ is the goal distribution. The objective is to obtain a policy $\pi(\mathbf{a_t}|\mathbf{s_t}, \mathbf{g})$ to maximize the expected sum of rewards $\mathbb{E}[\sum_t R(\mathbf{s_t}, \mathbf{g})]$, where the goal is sampled from $\rho_g$ and the states are sampled according to $\mathbf{s_0} \sim \rho_0$, and $\mathbf{s_{t+1}} \sim \mathbf{p(s_{t+1}|s_t, a_t)}$.

We define the state $s$ as the poses of $N$ objects in the scene. The actions $a \in A$ consist of the choice of action $\psi$, end-effector position $x$ and planar orientation $\theta$: $a = (\psi, x, \theta), \psi \in \{\text{PUSH}, \text{GRASP}\}, x \in \mathbb{R}^3, \theta \in \mathbb{R}$. We choose a sparse reward for actions - 1.0 for successful GRASP and 0.5 for successful PUSH. The higher reward for GRASP incentivizes the robot to prioritize it over PUSH when both are available. We consider a PUSH successful if the pixel-wise change in the depth image after a PUSH is larger than a pre-defined threshold. The intuition behind designing the PUSH reward this way is that we only use it for singulating objects in clutter where direct GRASP is not available, not for object rearrangement. A GRASP is considered successful if the antipodal distance between the parallel-jaw gripper fingers after a GRASP attempt is higher than a pre-defined threshold. Observation $o_t$ is defined as the RGB-D image captured by a statically mounted camera. The goal specification $o_g$ is the RGB-D image of the goal arrangement from the same camera viewpoint.

Given the current observation $o_t$ we use fully convolutional neural networks (FCNs) to model Q-functions that estimate the expected reward for each PUSH and GRASP candidate. The deep Q-learning framework is shown in Figure 3a. A 121-layer DenseNet [35] pretrained on ImageNet [36] is used to extract visual features from raw RGB-D images. In each FCN, we have three $1 \times 1$ convolutional layers; we apply batch normalization and ReLU activation before every convolutional layer. After FCN, we upsample with bilinear mode to have a pixel-wise Q-value estimate of the same size as input images. Each pixel unit in the Q-value map corresponds to the expected reward for executing an action at this pixel location. For each action we model end-effector orientation

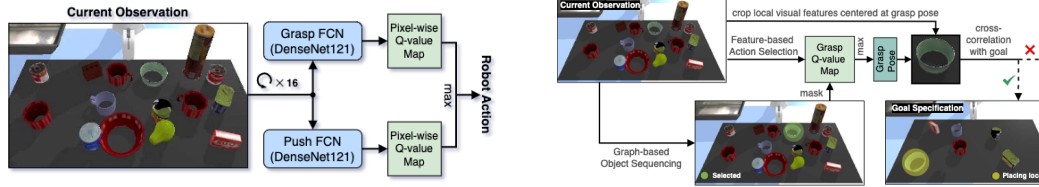

(a) Feature-based Action Selection      (b) Correspondence-based Reposition

Figure 3: **Subpolicies.** (a) A deep Q-learning framework maps the visual observations to actions, similar to [20]. (b) The grasped object placement is conditioned on the cross-correlation between the visual feature of the goal scene and the local features of the grasped object.

by rotating $o_t$ to 16 different orientations. Thus we have 32 pixel-wise Q-value maps (16 each for PUSH and GRASP). Each represents the Q-value estimate of executing the corresponding action at that orientation at all pixel locations. At each timestep $t$, before the robot chooses the next action, we mask all 16 GRASP Q-value maps with the output from the graph-based object rearrangement sequencing module (Figure 3b) to rule out objects that do not currently need to be repositioned. Following this, the robot picks an action (PUSH or GRASP) with the highest Q-value and executes it at the corresponding pixel location and end-effector orientation.

Loss is calculated by computing the temporal difference (TD) between the estimated reward and the actual obtained reward after execution. We only compute the loss for the selected pixel/pose (where the robot will take the next action); all other pixels/poses backpropagate with loss 0. We generate the label for PUSH at time $t$, $y_t^{\text{PUSH}}$, by calculating the depth image change after the push – if it is higher than a predefined threshold we consider the PUSH successful, $y_t^{\text{PUSH}} = 0.5$. For GRASP, we obtain the label at time $t$, $y_t^{\text{GRASP}}$, via the feedback signal from the gripper, if the antipodal distance between parallel jaws is larger than a predefined threshold, we consider the gripper is holding the object and hence the GRASP is successful, $y_t^{\text{GRASP}} = 1$. We use a Huber Loss for both PUSH and GRASP. For action executed at time $t$, let $y_t$ denote the label, $Q_t$ denote the estimated reward. The TD is given by $|Q_t - y_t|$, and the primitive learning loss is calculated as:

$$\mathcal{L}_p = \begin{cases} \frac{1}{2}(Q_t - y_t)^2, & |Q_t - y_t| < 1, \\ |Q_t - y_t| - \frac{1}{2}, & \text{otherwise.} \end{cases}$$

### 3.2 Correspondence-based Reposition: Where to PLACE

We model finding PLACE pose at time $t$ as a template matching problem [37] conditioned on the current observation $o_t$, the goal specification $o_g$, and the successfully executed GRASP $\tau_{t-1}$ at time $t-1$. We use a pretrained ResNet [38] to extract visual feature maps for both $o_t$ and $o_g$. Let $\phi(o_t)$ denote the visual feature map for $o_t$. Given the executed GRASP $\tau_{t-1} = (x_{t-1}, \theta_{t-1})$, where $x_{t-1}$ represents the GRASP location and $\theta_{t-1}$ represents the end-effector rotation, we crop a visual feature segment $\phi(o_{t-1})[\tau_{t-1}]$ with a predefined crop window size centered at $x_{t-1}$, and we consider $\phi(o_{t-1})[\tau_{t-1}]$ as a template for the grasped object (Figure 3b). The cross-correlation between $\phi(o_{t-1})[\tau_{t-1}]$ and $\phi(o_g)$ outputs a similarity distribution showing the resemblance between $\phi(o_{t-1})[\tau_{t-1}]$ and the local features at every placement in $\phi(o_g)$ given by $\varphi_t^{\text{similarity}} = \phi(o_{t-1})[\tau_{t-1}] * \phi(o_g)$.

Different from [37], we also apply cross-correlation between depth images $o_g^{\text{depth}}$ and $o_t^{\text{depth}}$: $\varphi_t^{\text{depth}} = \phi(o_t^{\text{depth}}) * \phi(o_g^{\text{depth}})$, which outputs a pixel-wise distribution over the workspace indicating whether a pixel location is occupied by objects in the current scene or in the goal scene. The prediction for PLACE is given by: $\varphi_t^{\text{PLACE}} = \varphi_t^{\text{similarity}} - \varphi_t^{\text{depth}}$. By lowering the value for occupied pixels we avoid placing the grasped object on top of other objects or at goal positions of other objects. $\varphi_t^{\text{PLACE}}$ is a pixel-wise prediction and each pixel represents a potential placement for the grasped object; to model the end-effector rotation of PLACE, we rotate the current image $o_t$ to 16 different orientations as input and pick the one with the highest prediction value. The non-occupied location in $\phi(o_g)$ with the highest cross-correlation value is considered as the best PLACE $\tau_t^{\text{place}}$ for the grasped object. If a match cannot be found in $o_g$, i.e. the similarity score is below a predefined threshold for all pixel locations, the object is placed aside in the bin. The training loss for PLACE policy learning is cross-entropy. The ground-truth goal position and orientation of the grasped object are extracted

directly from the simulator. We generate the label $y_t^{\text{PLACE}}$ by assigning value 1 to the pixel at the goal location of the grasped object; all other pixels are set to 0. The learning objective is to maximize the visual feature extraction model's prediction accuracy given a goal image and a template. While we rotate the input image in 16 different directions to differentiate placing orientations, only assign value 1 for the one with correct goal orientation. The rearrangement loss is calculated as:

$$\mathcal{L}_r = -y_t^{\text{PLACE}} \log \phi_t^{\text{PLACE}} + (1 - y_t^{\text{PLACE}}) \log(1 - \phi_t^{\text{PLACE}}).$$

### 3.3 Graph-based Object Sequencing: Which Object is Next

**Graph Generation**  We construct an accessibility graph representing reachable traversal paths from the robot end-effector location to every object. Unlike [39] (which assumes a known geometry for objects and uses the graph for target object retrieval tasks), we use UOIS-Net-3D [34] to provide a set of object segmentation masks from raw RGB-D images. We consider each segmented object as a vertex $v \in \mathcal{V}$ in the scene graph and add $v_r$ as the robot vertex. An edge $e \in \mathcal{E}$ between a pair of vertices means a collision-free end-effector path exists between them. The graph generation algorithm is shown in algorithm 1. Examples of generated scene graphs are in the supplement. The

---
**Algorithm 1:** ACC-GRAPH GENERATION
---
**Input:** camera observation $\mathcal{O}$ of a scene.
**Output:** accessibility graph $\mathcal{G} = (\mathcal{V}, \mathcal{E})$.
1   $\mathcal{E} \leftarrow \varnothing, \mathcal{V} \leftarrow \varnothing, \mathcal{V}' \leftarrow \varnothing$
2   *Get segmentation from UOIS-Net-3D($\mathcal{O}$)*
3   *Each segmented object maps to $v \in \mathcal{V}'$*
4   *Create robot vertex $v_r$, $\mathcal{V} \leftarrow \{v_r\}$*
5   **while** $\exists v \in \mathcal{V}'$ *and* $v \notin \mathcal{V}$ **do**
6     **for** *every $v_i \in \mathcal{V}$* **do**
7       **for** *every $v_j \in \mathcal{V}'$* **do**
8        **if** *linear distance path $(v_i, v_j)$ is collision-free* **then**
9         $\mathcal{E} \leftarrow \mathcal{E} \cup \{(v_i, v_j)\}$
10         $\mathcal{V} \leftarrow \mathcal{V} \cup \{v_j\}$
11         $\mathcal{V}' \leftarrow \mathcal{V}' - \{v_j\}$
12   **return** $\mathcal{G} = (\mathcal{V}, \mathcal{E})$
---

traversal path from the robot vertex to an object vertex in the generated scene graph captures the shortest path from the robot base that object including objects blocking the straight line path.

**Object Sequencing**  Let $o_t$ and $o_g$ denote the current and the goal scenes, and $\mathcal{G}_t$, $\mathcal{G}_g$ denote the current and the goal scene graphs. We establish a list of sub-graphs of $\mathcal{G}_t$ by individually removing each vertex and its related edges. We calculate the similarity between each sub-graph and the goal graph through a pretrained SimGNN [40], previously shown to be an excellent approximator ($MSE < 1.18 \times 10^{-3}$). The graph similarity corresponds to the Graph Edit Distance (GED) between graphs $G_1$ and $G_2$ – the number of edit operations in the optimal alignment that transform $G_1$ into $G_2$, where an edit operation on a graph is an insertion or deletion of a vertex/edge or relabelling of a vertex (isomorphic graphs have GED 0). The removed vertex (object) from the highest similar-

---
**Algorithm 2:** OBJECT REARRANGEMENT SEQUENCING
---
**Input:** accessibility graphs of the current and the goal scene, $\mathcal{G}_t = (\mathcal{V}_t, \mathcal{E}_t)$ and $\mathcal{G}_g = (\mathcal{V}_g, \mathcal{E}_g)$.
**Output:** selected object $v \in \mathcal{G}_t$ for next rearrangement.
1   $n \leftarrow \mathcal{V}_t.size$
2   *Initialize an array* $\text{sim}[1, ..., n] \leftarrow 0$
3   **for** *every $v_i \in \mathcal{V}_t$* **do**
4     $\mathcal{G}_t^i \leftarrow \mathcal{G}_t - \{v_i\}$
5     $\text{sim}[i] \leftarrow \text{Sim\_GNN}(\mathcal{G}_t^i, \mathcal{G}_g)$
6   $selected \leftarrow \arg \max \text{sim}[1, ..., n]$
7   **return** $\mathcal{V}_t[selected]$
---

ity sub-graph is selected to be rearranged next. The robot thus chooses the object responsible for the largest difference between the current and goal scene graphs, to keep the number of task completion actions as low as possible. We show the process of choosing next object to rearrange in algorithm 2.

**Loss Calculation**  We use $A^*$ to calculate the ground-truth GED between graphs [41], since the scene graphs are small. To scale up to complex graphs in the future (where the ground-truth GED might be inaccessible or computationally expensive to obtain) we use SimGNN to approximate GED for all graphs instead of using the ground-truth GED. We transform the ground-truth GED between $G_1$ and $G_2$ to ground-truth similarity labels $y$ in the range $(0, 1]$ [40]:

$$y = e^{-\text{Norm\_GED}(G_1, G_2)} \qquad \text{Norm\_GED}(G_1, G_2) = \frac{\text{GED}(G_1, G_2)}{(|G_1| + |G_2|)/2}$$

where $|G|$ is the number of vertices in $G$. Let $s_i$ denote the similarity prediction output between $\mathcal{G}_t^i$ and $\mathcal{G}_g$ from SimGNN and $y_i, i = 1, ..., N$ denote the ground-truth similarity label. We use the cross-entropy loss for the graph-based object rearrangement sequencing:

$$\mathcal{L}_s = -\sum_{i=1}^{N} [y_i \log s_i + (1 - y_i) \log(1 - s_i)].$$

After selecting an object for rearrangement, its placement is located as described in subsection 3.2.

# 4 Evaluation

## 4.1 Experimental Results in Simulation

We use a position controlled Franka Panda arm with a parallel-jaw gripper in Pybullet [42]. A simulated RealSense D415 RGB-D camera with resolution $640 \times 480$ is statically mounted. A side bin is placed to hold objects removed from the workspace.

| Method | Rotation | Swap | Clutter | Selectivity | Init. #obj. | Goal #obj. | Completion ↑ | Position Error ↓ |
|---|---|---|---|---|---|---|---|---|
| NeRP [24] | ✗ | ✔ | ✗ | ✗ | 3-8 | 3-8 | 94.56 ± 0.73 | 1.90 ± 1.30 |
| IFOR [25] | ✔ | ✔ | ✗ | ✗ | 1-9 | 1-9 | 81.80 | 2.70 ± 2.30 |
| | ✔ | ✗ | ✗ | ✗ | 3-7 | 3-7 | 96.67 ± 1.67 | 1.29 ± 0.91 |
| | ✔ | ✔ | ✗ | ✗ | 3-7 | 3-7 | 90.00 ± 3.00 | 2.99 ± 2.37 |
| Ours | ✔ | ✗ | ✗ | ✔ | 3-7 | 1-5 | 97.33 ± 0.67 | 1.41 ± 2.70 |
| | ✔ | ✔ | ✗ | ✔ | 3-7 | 1-5 | 97.00 ± 1.00 | 1.81 ± 2.66 |
| | ✔ | ✗ | ✔ | ✔ | 16-20 | 5-10 | 85.67 ± 2.33 | 1.64 ± 0.44 |
| | ✔ | ✔ | ✔ | ✔ | 16-20 | 5-10 | 82.33 ± 2.67 | 1.22 ± 0.93 |

Table 2: **Task Completion (mean %) and Position Error** ($10^{-2}m$). Init./Goal #obj. is the number of objects in the initial/goal scene respectively. NeRP [24] and IFOR [25] are state-of-the-art models for image-based tabletop rearrangement. Statistics quoted here are from their original paper (codebases not publicly available). NeRP is restricted to translation. Since we handle both translation and rotation we compare with IFOR. We achieve a higher task completion (90%) than IFOR (81.8%) in the same setting (reduced clutter, no selectivity, but swaps may be needed). We handle significantly more complex cases than either previous model (e.g., last row of the table shows concurrent challenges handled by our system: high clutter, selectivity, with swaps needed).

We conduct 6 sets of simulations (3 random seeds each, 100 episodes) with variations shown in Table 2. In each episode, we randomly pick $3 \leq N \leq 20$ YCB objects [43] and select a subset (or all $N$) to rearrange. Objects to be rearranged are placed at random goal positions and orientations and an RGB-D image is captured. This image is the goal specification. Next, we randomly reposition and rotate all these objects and add the remaining objects (those not designated for rearrangement) at randomly generated positions and orientations. The resulting scene is the initial state of the episode.

We add objects in testing that were not seen in training to show the system's adaptability to novelty. We differentiate task difficulty by measuring the degree of scene clutter, the degree of selectivity (how many of the objects are designated for rearrangement), and how many swap actions are needed. Let $\mathbf{P} = \{(x_1, y_1), ..., (x_n, y_n)\}$ denote object positions. We define a **clutter** coefficient:

$$c(\mathbf{P}) = -\log\left\{\frac{1}{n}\sum_i^n \sqrt{(x_i - \hat{x_i})^2 + (y_i - \hat{y_i})^2}\right\}, \quad \hat{x_i} = \mathbf{kNN}(\mathbf{y}), \hat{y_i} = \mathbf{kNN}(\mathbf{x}),$$

in which $\mathbf{kNN}(\mathbf{y}), \mathbf{kNN}(\mathbf{x})$ estimates $\hat{x_i}, \hat{y_i}$ through k-nearest neighbor regression on every other object's position. We consider arrangements with $c(\mathbf{P}) \geq 1.0$ as 'cluttered' (examples in supplement Appendix A). In **selective** episodes, a proper subset of initial objects is in the goal arrangement. Hence, the system must identify which objects need to be manipulated. Some episodes require **swaps**, where the goal positions of certain objects are initially occupied by others. This requires the robot to first move the blocking object and then reposition the original object to be rearranged.

We evaluate our method with three metrics: **(1) Task completion** is the percentage of completion in all rearrangement episodes. We consider an episode to be complete when all target objects are placed within 5 cm from their goal position (consistent with [25]) and all non-target objects are placed aside. **(2) Position error** is the average Euclidean distance between the desired arrangement and the final arrangement achieved. **(3) Planning steps** is the average number of actions the robot takes in each completed episode. It is a measure of the planning efficiency of the learned rearrangement policy.

Our system performs rearrangement in a variety of settings, generalizing readily from 3-20 objects (Table 2, Figure 4). Task completion is calculated over all test episodes; planning steps and position error are only reported on successful episodes. The task completion rate decreases as the task

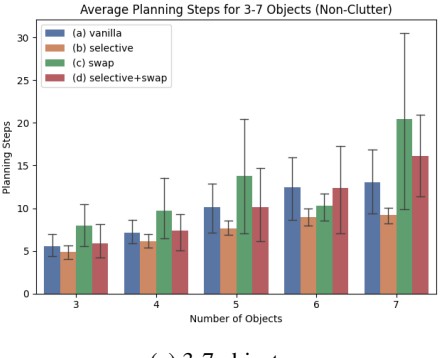
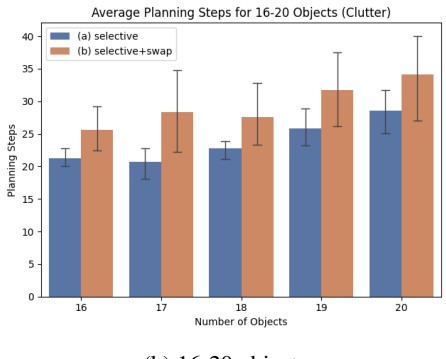

(a) 3-7 objects                                                                 (b) 16-20 objects

Figure 4: **Average Planning Steps** Tasks with target selection require fewer planning steps, introducing swap actions in the task setting increases planning steps. The number of planning steps increases as the number of objects in the scene grows.

setting becomes more difficult, the position error is stable across all task settings, which indicates an accurate placement prediction from the sub-policy shown in Figure 3b. In selective episodes, our system has a higher task completion rate than non-selective episodes when other task settings are the same. We ascribe this to the graph-based object sequencing module (subsection 3.3) that prioritizes removing non-target objects over rearranging target objects, thus decreasing the clutter coefficient of the current scene and potentially improving the success rate (see supplement Table 2 for details). The task completion rate decreases in situations with high clutter and swap actions. With increased clutter, it is more difficult to find 'buffer' locations for objects whose goal positions are occupied by other objects or objects that are occupying others' goal locations.

NeRP [24] and IFOR [25] are state-of-the-art models for image-based tabletop rearrangement. Like IFOR, our method includes planar rotation alignment of objects (examples in supplement subsection B.2) while NeRP only considers translations. Compared to IFOR we achieve a 8.2% higher task completion rate in the same task setting at a comparable rotation error (ours:13.89°, IFOR:13.70°).

In Figure 4, we observe that when the task setting remains the same, the number of planning steps increases as the number of objects increases. When target selection is involved, the number of planning steps decreases, as the object sequencing mechanism prioritizes removing non-target objects from the table, leaves a more sparse arrangement of objects in the workspace, potentially reducing subsequent task difficulty. The introduction of swap actions, however, significantly increases the number of planning steps in each task completion. The swap action requires the robot to sample 'buffer' locations for objects whose goal position is occupied, place objects at 'buffer' locations, remove the 'placeholder' objects at their goal positions and then reposition the objects at their goal locations. This process naturally adds more required actions towards task completion.

Two noteworthy recent rearrangement systems are TRLB [10] and ReorientBot [26]. Both rely on suction mechanisms to manipulate objects in clutter without the need to singulate them. TRLB relies on the initial and goal states being fully specified as object poses with a focus on fast planning for rearrangement and ReorientBot relies on the goal state being fully specified as object poses. Our task is sufficiently different (gripper instead of a suction mechanism, goal specified only by a single image) making a direct comparison between our work and these two systems infeasible.

## 4.2 Ablation Studies

**Target Object Selection:** In selective rearrangement, the objects in the goal scene (target objects) might be a subset of those in the initial scene. We evaluate the significance of using ResNet to obtain an accurate visual feature cross-correlation and target object classification by testing 2 different encoder-decoder structured visual feature extractors, ResNet [38] and U-Net [44]. We measure the match success rate, average position prediction error and target object classification accuracy over 100 different initial and goal arrangements. The choice of visual feature extraction model is crucial because it directly affects the accuracy of target object identification and reposition. Experiments show that ResNet achieves match success rate of 93.33%, position error $<2.04$ cm and target classification accuracy of 98.58% with 1-20 objects (details in supplement subsection B.3).

**Graph-based Object Sequencing:** To verify the importance of graph-based object sequencing to minimize the number of actions, we test two scene graph generation methods and measure their impact on the average number of planning steps. We also consider the situation when no sequencing mechanism is used (no scene graph) and the robot picks the next object only based on PUSH and

| Scene Graph | 10 | 20 |
|---|---|---|
| N/A | 35.13±3.55 | 45.22±4.70 |
| Position | 19.94±4.93 | 29.29±3.52 |
| Accessibility | **15.61**±3.84 | **25.45**±3.88 |

Table 3: **Scene Graph Comparisons.** Average planning steps vs. # of objects in the initial scene. All scenarios have 10 target objects.

GRASP Q-value estimates. We generate the scene graph in two ways; a position-based approach which captures the basic spatial relationships among objects and an accessibility approach (subsection 3.3 algorithm 1). We perform object sequencing (algorithm 2) given the scene graph $\mathcal{G}_t$ and the goal scene graph $\mathcal{G}_g$. Compared with no sequencing, using the accessibility graph decreases planning steps by 55.56% (10 objects) and by 43.71% (20 objects). Compared with the position-based approach, using the accessibility graph decreases planning steps by 21.72% (10 objects) and by 13.11% (20 objects) thus confirming the efficacy of graph-based object sequencing. A detailed analysis is in the supplement subsection B.4.

### 4.3 Demonstration on a Physical Robot

We test our system on a Panda robot arm with a parallel-jaw gripper, and a statically-mounted RGB-D camera overlooking the tabletop (Figure 5). A bin next to the workspace holds the redundant (non-target) objects. Objects included in the demonstration vary across experiments, including a collection of 20 daily use objects (e.g. peanut butter jar, ketchup bottle). The robot demonstration generalizes to novel objects not available during training.

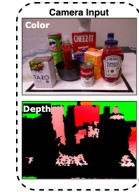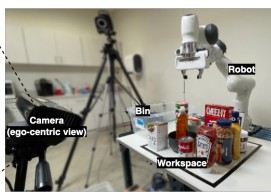

Figure 5: **Robot experiments.**

We show zero-shot transfer from simulation to a real robot setting in the video.

## 5 Limitations

Our system has several limitations. **(1) 6 DOF rotation.** We are limited to planar object rotations. We do not currently handle 6 DOF object reorientation, and our system is poor at orienting rotationally symmetric natural objects (*e.g.,* oranges). **(2) Cluttered final state.** Even though our method solves difficult rearrangement tasks with cluttered *initial* object arrangements, it struggles with scenarios where the desired goal arrangement is cluttered. Unsurprisingly, this is a significant challenge for other existing systems too – with a large number of objects it turns into a difficult packing or stacking problem. **(3) Segmenting objects.** Our system is object-centric – we use scene segmentation to build scene graphs and sequence objects. Incorrect object segmentation produces inaccurate object sequencing and performance degradation. **(4) Underlying motion planner limitations.** In some experiments, we experienced difficulties with joint limits being reached when the initial grasp for an object turns out to not be feasible for object placement in the new location or when the robot carrying an object collides with another object. We believe limitations 1,3 and 4 can be addressed respectively by expanding the action space in the action selection module, better/multiple cameras, and improved image segmentation techniques and trajectory-aware obstacle-avoiding planners.

## 6 Conclusions

We proposed an effective image-based learned method for selective tabletop object rearrangement in clutter. Our simulated experiments provide evidence that the method works well in challenging settings which require the robot to begin with an intially cluttered scene, select only the objects that need to be rearranged while discarding others, deal with cases where the target location for an object is already occupied - making the system the first of its kind to be able to address all these concurrently. Ablation studies provide an analysis of system performance. We also demonstrate zero-shot transfer of our system to a real robot and generalization to unseen objects.

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
