# OpenReview forum: "Selective Object Rearrangement in Clutter"
_robot-learning.org/CoRL/2022/Conference — CoRL 2022 Poster_

### Official Review · Reviewer_Z5Z1 · 2022-07-26

**Originality:** Good
**Technical Quality:** Good
**Clarity Of Presentation:** Very Good
**Impact:** 3

**Recommendation:**

Weak Reject: I recommend rejecting the paper, but will not argue for my recommendation if the majority of other reviewers have a different opinion.

**Summary:**

This paper presents a system for image-based object rearrangement in cluttered tabletop environments where given a goal image, the robot plans a sequence of grasp and pushing actions to achieve the configuration corresponding to the goal image. The presented approach has 3 components: (1) an object sequencing method, which constructs a scene graph and encourages the robot to manipulate objects which  result in a graph closer in topology to the scene graph corresponding to the goal image, (2) an action selection method, which selects a grasping or pushing action on sequenced objects based on their likelihood of success, and (3) a placement module which places objects such that they match their pose in the goal image. The action selection method and placement module are very similar to ideas from prior work, but the object sequencing method appears to be relatively novel. The constructed system is also impressive: the videos demonstrate the efficacy of the approach on robot hardware.

**Issues:**

Major issues:

1) Please clearly describe in more detail why prior TAMP methods with learning-based vision models [14, 15, 16, 17] would not scale to the scenarios evaluated in this paper and ideally provide experimental evidence for this argument. Without this, it is difficult to understand why the presented algorithm is needed with respect to prior existing work.

2) It would be good to implement some prior algorithms as discussed in the weaknesses above in simulation to provide a calibrated comparison to prior work. Currently, there does not seem to be concrete experimental evidence that the proposed approach provides significant benefit over prior approaches.

3) Please report full quantitive details of physical experiments as indicated in the weaknesses section above.

More minor issues, but which would benefit from clarification:

4) Why are pushes considered successful based on pixel change in the image? If the purpose of a push is to increase grasp access, shouldn't pushes be considered successful based on the degree to which the increase grasp Q-values?

5) The grasp success criteria is a bit strange. Just because the antipodal distance between parallel jaws is larger than a predefined threshold doesn't seem to necessarily suggest that the executed grasp will be stable when the object is lifted. More clarification on this point would be much appreciated.

6) The learning objective in Section 3.2 is somewhat confusing. Specifically, I am confused by how $\varphi_t^{\text{PLACE}}$ and $\phi_t^{\text{PLACE}}$ are related to each other. Given $\varphi_t^{\text{PLACE}}$ (which requires no learning to compute since a freezed Resnet backbone is used to featurize images), it seems like all you would need to do is place the object at the location where $\varphi_t^{\text{PLACE}}$ is the highest.

7) I am also confused by the learning objective in Section 3.3. The authors initially state that a pre-trained SimGNN is used to determine graph similarity. Given this, I am confused why any learning is necessary at all (and am thus confused by the loss definition $\mathcal{L}_s$: given this graph similarity measure, you have all the information you need to determine which object to sequence next.



**Quality Of The Limitations Section:**

Limitations are addressed clearly

**Reviewer Expertise:**

4: The reviewer is confident but not absolutely certain that the evaluation is correct

**Robotics Focus:**

Sufficient demonstration on hardware

**Strengths And Weaknesses:**

Strengths:

(1) The presented method is clearly described and for the most part all components of the system are easy to understand.
(2) The resulting system is deployed on physical robotic hardware and videos suggest that the proposed method is able to successfully rearrange configurations with up to 15 objects. This is a very challenging task and the effort put into deploying this system in hardware experiments is much appreciated.
(3) The graph based object sequencing method is very interesting and novel to the best of my knowledge. I appreciate the ablation in Section 4.2, which appears to demonstrate the utility of the approach as opposed to heuristics based only on grasp Q-values (such as in Mech Search by Danielczuk et al.)
(4) I appreciate the extensive evaluation of the proposed method on a number of different numbers of initial objects and goal objects in simulation.

Weaknesses:

(1) The novelty of the work with respect to prior work needs to be made a bit more clear. The paper claims that this is the first paper that allows the robot to selectively rearrange objects from image observations in clutter. While the authors claim that prior TAMP approaches that incorporate learning-based vision models [14, 15, 16, 17] struggle to scale to adversarial environments, the reason for this should be expanded in more depth, and preferably, the authors should experimentally show that prior methods which seem in principle capable of addressing the same problem setting are intractable for the problem settings considered in this work.

(2) The comparison to prior work in simulation experiments does not appear calibrated. The authors should implement prior methods in the same setup as their algorithm is evaluated in. Otherwise, the numbers in Table 2 do not seem to be a fair comparison, and it is hard to draw any meaningful conclusions about the relative performance of the presented method. It seems important to implement at least 2 of [14, 15, 16, 17, 24, 25] based on which the authors believe will be the strongest baselines in their experimental setup.

(3) The discussion of the physical experiment results is insufficient in the main body of the paper. It is important the paper discusses quantitive results such as the number of trials, success rate, etc... of these experiments for the reader to be able to evaluate the significance of the results. It is also preferable to compare to prior methods in physical experiments, but it is OK to omit this if sufficient comparison is at least provided in simulation.

**Summary Of Recommendation:**

This paper presents an impressive robotic system for object rearrangement with convincing evaluation on physical hardware. However, the experimental evaluation with respective to prior work and reporting of results in physical experiments need substantial improvement to fully evaluate the contributions of the paper. I am happy to update my recommendation if these issues are sufficiently addressed, but I am currently learning towards rejection for these reasons.

---

> ### Author Response · Authors · 2022-08-22
> **Response to Official Review of Paper147 by Reviewer Z5Z1**
>
> **Comment:**
>
> Please note that our response has been split into two parts due to the space constraints. (Part 1/2)
>
> We thank the reviewer for constructive comments. We are pleased that you found our paper well organized and clear.
>
> **Major issues:**
>
> - Please clearly describe in more detail why prior TAMP methods with learning-based vision models [14, 15, 16, 17] would not scale to the scenarios evaluated in this paper and ideally provide experimental evidence for this argument. Without this, it is difficult to understand why the presented algorithm is needed with respect to prior existing work.
>
> This is a great point! In this work, we proposed a framework that can solve extremely cluttered tabletop rearrangements. For highly-cluttered tabletop, direct grasp of an object might not be accessible. A simple example of such cases is shown in Figure 1 in supplementary material where the clutter coefficient is larger than one. When the gripper is not wide enough to grasp more than one object, no grasp is available if the objects are not separated from the clutter first. We introduced push as an additional action primitive in our system for the purpose of separating objects when no direct grasp available in dense clutters. To our best knowledge, no previous object rearrangement systems include push actions in their action space to benefit grasp success in dense clutters.
>
> [14] uses goal poses as goal specification while our system directly uses a single image. Also, they plan the action sequence from the initial scene image, which can be really challenging when the number of objects in the scene increases (clutter) and requires replanning when new object instances are revealed during rearrangement (occlusion). This is also mentioned in the last section of their paper.
>
> [15] represents procedures using PDDLStream which provides structural constraints. Their planning is based on a set of predicates, which introduces the tradeoff between the number of objects included in the system and planning/replanning efficiency and relies on a structured environment.
>
> [16] depends on the knowledge of the environment. As indicated in the last section of their paper, when the environment is dynamic and not fully observable, the horizon of their planning is limited. While our system only relies on the current camera data and the given goal image (Markovian), it can adapt to dynamically changing and partially observable environments.
>
> [17] assumes universal shapes of all objects (round collision radius) which does not apply when object shapes and sizes vary like everyday objects in YCB dataset.
>
> - It would be good to implement some prior algorithms as discussed in the weaknesses above in simulation to provide a calibrated comparison to prior work. Currently, there does not seem to be concrete experimental evidence that the proposed approach provides significant benefit over prior approaches.
>
> We did not reimplement the previous state-of-the-art object rearrangement systems ([24] with translation only and [25] when rotations are considered) because without access to their codebase and other software/hardware configuration details, we found it risky to replicate their systems when there are numerous implementation details (such as physical robot setting, lighting, hyperparameters, training time, random seeds, and hardware) that can largely influence the overall performance of the system. However, we tried to mirror their experimental setting as close as possible in order to provide a calibrated and informative comparison between our system and previous works:
> 1. Same input format: images of the current scene and single image of the goal scene.
> 2. Same data generation process: randomly place everyday objects (from YCB dataset) on an open tabletop, the number of objects involved are consistent with previous works.
> 3. Same evaluation matrices: task completion, planning steps and position error.
>
> - Please report full quantitative details of physical experiments as indicated in the weaknesses section above.
>
> We only did a demonstration of our system on a physical robot with a small number of trials where we do not feel fair to report the statistics on such a small set of experiments.
>
>
>
> **Zip File:**
>
> /attachment/2e909fabe9eb840adf3b0ea86a8d275b54f91c72.zip

---

> > ### Comment · Reviewer_Z5Z1 · 2022-08-24
> > **Thank You for YOur Response**
> >
> > It is critical to actually reimplement previous state-of-the-art methods to the extent possible with the experimental setup calibrated with your own. Without this, I cannot draw anything meaningful from the comparisons reported to these approaches. I also still don't understand why clear details on the physical experiments are not reported: even if there are a small number of trials, you should fully report the result of those trials. As currently reported, the physical experiments do not carry much weight in my evaluation because they are not rigorously conducted or properly reported. As a result my evaluation remains unchanged (Weak Reject).

---

> ### Author Response · Authors · 2022-08-22
> **Response to Official Review of Paper147 by Reviewer Z5Z1**
>
> Please note that our response has been split into two parts due to the space constraints. (Part 2/2)
>
> **Minor issues:**
>
> - Why are pushes considered successful based on pixel change in the image? If the purpose of a push is to increase grasp access, shouldn't pushes be considered successful based on the degree to which the increase grasp Q-values?
>
> We introduce push actions to deal with cluttered scenarios, push actions are essentially intended to separate objects. We believe the change in the scene (i.e. pixel changes in the image) indicates the change in object arrangement, and increases the possibility for grasp success. In other words, push actions are only executed when no grasp action is available (i.e. the scene is cluttered), so by causing difference in the scene, will further enable future grasps.
>
> - The grasp success criteria is a bit strange. Just because the antipodal distance between parallel jaws is larger than a predefined threshold doesn't seem to necessarily suggest that the executed grasp will be stable when the object is lifted. More clarification on this point would be much appreciated.
>
> We measure the antipodal distance between parallel jaws after the robot executes the lift action. The reviewer is correct - this is not a guarantee the grasp is the most stable, but we see in practice that it is a useful proxy for whether the current object is successfully lifted (and hence a successful grasp).
>
> - The learning objective in Section 3.2 is somewhat confusing. Specifically, I am confused by how and are related to each other. Given (which requires no learning to compute since a freezed Resnet backbone is used to featurize images), it seems like all you would need to do is place the object at the location where is highest.
>
> It does not require learning to compute, however, the learning objective here is to “finetune” the network so it captures the visual similarity as accurately as possible.
>
> - I am also confused by the learning objective in Section 3.3. The authors initially state that a pre-trained SimGNN is used to determine graph similarity. Given this, I am confused why any learning is necessary at all (and am thus confused by the loss definition given this graph similarity measure, you have all the information you need to determine which object to sequence next.
>
> Pre-trained SimGNN is trained on high-dimensional graph data like AIDS, LINUX, IMDB dataset which gives us a good approximate of graph similarity while we add the training here to future finetune the network to adapt to our generated scene graph, in order to improve system performance.

---

### Official Review · Reviewer_W89x · 2022-07-31

**Originality:** Very Good
**Technical Quality:** Very Good
**Clarity Of Presentation:** Very Good
**Impact:** 4

**Recommendation:**

Weak Accept: I recommend accepting the paper, but will not argue for my recommendation if the majority of other reviewers have a different opinion.

**Summary:**

Selective Object Rearrangement in Clutter presented an image-based learned method for table-top object rearrangement in clutter using parallel jaw gripper. The method consists of three stages, including a graph-based object sequencing to select the next object to manipulate (using the current scene graph and goal scene graph's graph edit distance), feature based action selection to map image to robot actions (select between pushing or grasping), and a correspondence-based placement policy. Overall, authors claim that the method is the first to address concurrently object selection from initially cluttered scene, discarding the unrelated objects, and dealing with cases where goal location is occupied. Experiments show that the method is able to outperform SOTA methods in less restrictive settings, while also providing reasonable performance in more restrictive settings.

**Issues:**

See strengths and weaknesses section.

**Quality Of The Limitations Section:**

Limitations are addressed clearly

**Reviewer Expertise:**

3: The reviewer is fairly confident that the evaluation is correct

**Robotics Focus:**

Sufficient demonstration on hardware

**Strengths And Weaknesses:**

Strengths
* Real robot experiments: The authors demonstrated zero-shot transfer from simulation to real robot. Cool!
* Interesting choice of actions: Push & Grasp, where grasp is preferred, push designed to singulate objects in clutter.
* Experiments are well designed, including definition of a clutter coefficient. Quantitative results of experiments analyzed, and convincing justifications given for observed behavior.
* Outperforms existing SOTA methods.

Weaknesses
* Pre-defined assumptions: Pre-defined window crop size for visually defining grasped object, would be better to see some ablations on generalization to out-of-distribution objects, would the assumption still hold?
* Dependence on pre-trained methods: UOIS-NET-3D is used to provide set of object segmentation masks. How does the system react to errors in the predicted segmentation mask?
* Missing experiment: It appears the NeRP baseline has adequate performance when only Swap is handled. It would be interesting to see how the presented method works when only swap is handled as well.
* Can the authors comment on whether this is an online method, and what kind of computational resources is required to run the pipeline?

**Summary Of Recommendation:**

The paper presented a novel comprehensive method tackling selective object rearrangement in clutter. Overall, the technical contributions are sound and clearly explained, with a full suite of experiments (simulation and real-robot). Through the reported results, it appears that the method outperforms existing baselines by a large margin, and is able to be scaled to real-world experiments zero-shot. There are a few points that I am hoping the authors can comment on / provide experiments for as outlined in the strengths & weaknesses section, including an evaluation of the method's dependence on pre-trained methods robustly performing, impact of pre-defined assumptions (i.e. fixed window crop size for template matching), and more careful comparison with NeRP.

---

> ### Author Response · Authors · 2022-08-22
> **Response to Official Review of Paper147 by Reviewer W89x**
>
> **Comment:**
>
> Thank you for your detailed and insightful comments! We are pleased to hear that you found our study to be extensive in simulated evaluations and that you agree that our work would be valuable for the community. You brought up several great points that we would love to discuss further below. Please let us know if further clarification is needed.
>
> **Weaknesses:**
>
> - Pre-defined assumptions:
>
> We believe the assumption still holds. The object selection in our evaluation varies largely, the fixed window crop size will only capture part of an object. However, since we are only comparing visual similarity, as long as the object appearance does not change drastically, the fixed crop size is not a major bottleneck for system performance.
>
> - Dependence on pre-trained methods:
>
> As mentioned in the limitation section, poor object segmentation will lead to system performance degradation. As the number of objects in the scene increases (more cluttered), UOIS-NET-3D gets less accurate which aligns with the results in Table 2, scenarios with more objects have worse performance compared to less object ones when all other parameters are the same.
>
> - Missing experiment:
>
> We have performed a new experiment with swap only. Our system handles both translation and rotation while NeRP is restricted to translation in object repositioning, which is demonstrated qualitatively in [25]. Despite this handicap, our system produces swap-only results that are comparable to NeRP.
>
> We have added the new results in the supplemental material pdf.
>
> - Can the authors comment on whether this is an online method, and what kind of computational resources is required to run the pipeline?
>
> Our method is online and it runs on a single RTX 3080 GPU.
>
> **Zip File:**
>
> /attachment/aba9167b18e23b097ea01963c6afcbb1d6b13e47.zip

---

> > ### Comment · Reviewer_W89x · 2022-08-28
> > **Thank you for the clarifications and added experiments**
> >
> > I appreciate the detailed responses that the authors have provided in addressing my earlier points (and points that fellow reviewers have raised). While based on the comparison with NeRP, there is a slight performance degradation for the swap only scenario, I believe that the ability for the system to handle both translation and rotation is more important. Overall, there could be some interesting extensions to this work (i.e. replacing template matching with a scene graph), looking forward to it. Since all of my points have been addressed, the idea is novel and interesting, and the experiments and ablations are quite extensive, I would like to raise my recommendation from a weak accept to a strong accept.

---

### Official Review · Reviewer_Exub · 2022-08-01

**Originality:** Very Good
**Technical Quality:** Excellent
**Clarity Of Presentation:** Very Good
**Impact:** 4

**Recommendation:**

Strong Accept: I recommend accepting the paper and will argue for my recommendation even if other reviewers hold a different opinion.

**Summary:**

This work presents a novel method for tabletop object rearrangement amid clutter for cases where a subset of the objects may need to be removed from the scene. It works through sequencing the objects to manipulate through a measure of the graph edit distance from the current state to the goal state, where each object is manipulated by either a learned grasp or a learned push primitive (both implemented as fully convolutional Q-value maps), and where each grasped object is either moved away from the scene or placed in a location with the greatest feature correspondence to the goal scene. Extensive evaluation is done in simulation, and it is demonstrated that the learned policies can successfully transfer to a real robot.

**Issues:**

See my summary of weaknesses - I feel the paper could still be improved with fairly substantial re-writing and iteration of the figures.

**Quality Of The Limitations Section:**

Limitations are addressed clearly

**Reviewer Expertise:**

5: The reviewer is absolutely certain that the evaluation is correct and very familiar with the relevant literature

**Robotics Focus:**

Sufficient demonstration on hardware

**Strengths And Weaknesses:**

Strengths:
* Clearly written and largely easy to follow
* The problem addressed is important to the field of robotics
* The proposed solution is novel in several respects (in particular the scene-graph edit based object sequencing and the depth-image cross correlation aspect of the feature-based repositioning).
* The experiments are detailed and convincingly demonstrate the approach works well, and the video clearly demonstrates the approach can be adapted to work on a real robot
* The discussion of limitations is thorough

Weaknesses:
* Several of the figures (1, 3, 4) have images and/or text that are so small that even zooming in does not make it easy to make out the details in the PDF, and that would be hard to understand in a printed version.
* I felt some of the text not covering the main novel ideas or results could have been cut or moved to allow for more results or analysis to be included in the main paper instead of the appendix. In particular, the stanard RL notation preliminaries are covered in section 3.1 and the summary of the Q-value map method (which as stated has a lot of overlap with [20]) to have been moved to the appendix or cut down. I feel including the planning step and ablation results in the main text would have been more informative.
* While the citation of the relevant prior work is well done, there is very little summary of it. I feel that for readers less familiar with the cited works would hard a hard time understanding which are the key novel contributions of this work and which parts are largely similar to prior work (such as the Q-value map).
* While the ablation results for the action sequencing is informative, I was left wondering how well a simple hand-coded heuristic (such as moving each non-target object to the bin first, and then repositioning each object by Q-value) might perform. In general, it is not intuitive why the first set of actions would not be just removing all non-target objects from the scene, followed by planning to reposition the remaining objects to the target.
* The goal specification including an image of the final scene (needed for template matching) seems a bit limiting -- the method would be more powerful and widely applicable were it able to work with just a scene-graph based of the goal scene.
* In some parts the paper can be a bit dense with extremely long paragraphs, and it seemed that it could have been edited a bit more to make reading easier.

**Summary Of Recommendation:**

While I feel this paper could still be improved in various ways, I still think the clear presentation of ideas, novel approach to an important problem, and impressive results strongly merit it being accepted.

---

> ### Author Response · Authors · 2022-08-22
> **Response to Official Review of Paper147 by Reviewer Exub**
>
> **Comment:**
>
> Thank you for your detailed and insightful comments! We are pleased to hear that you found our study to be extensive in simulated evaluations and that you agree that our work would be valuable for the community. You brought up several great points that we would love to discuss further below. Please let us know if further clarification is needed.
>
> **Weaknesses:**
>
> - Several of the figures (1, 3, 4) have images and/or text that are so small that even zooming in does not make it easy to make out the details in the PDF, and that would be hard to understand in a printed version.
>
> We have updated Figure 1, 3, 4 in the paper pdf.
>
> - I felt some of the text not covering the main novel ideas or results could have been cut or moved to allow for more results or analysis to be included in the main paper instead of the appendix. In particular, the standard RL notation preliminaries are covered in section 3.1 and the summary of the Q-value map method (which as stated has a lot of overlap with [20]) to have been moved to the appendix or cut down. I feel including the planning step and ablation results in the main text would have been more informative.
>
> We totally agree with this. We have shortened the main text to some extent already, and have included ablation results in the main text.
>
> - While the citation of the relevant prior work is well done, there is very little summary of it. I feel that for readers less familiar with the cited works would have a hard time understanding which are the key novel contributions of this work and which parts are largely similar to prior work (such as the Q-value map).
>
> Thank you for the suggestions. We have added a summary to the related work section.
>
> - While the ablation results for the action sequencing is informative, I was left wondering how well a simple hand-coded heuristic (such as moving each non-target object to the bin first, and then repositioning each object by Q-value) might perform. In general, it is not intuitive why the first set of actions would not be just removing all non-target objects from the scene, followed by planning to reposition the remaining objects to the target.
>
> In our framework, we use SimGNN for object sequencing, where the network almost always chooses to remove non-target objects first before manipulating target objects since they cause the most difference between the current and the goal scene graph. So the simple hand-coded heuristic approach will achieve similar performance with our approach in most cases. However, in scenarios where the workspace is extremely cluttered and some of the non-target objects are not accessible for grasp while some target objects are available, such approach might fail to complete the task or need more planning steps while the learning-based object sequencing combined with the feature-based action selection will choose to push or move the available target object first.
>
> - The goal specification including an image of the final scene (needed for template matching) seems a bit limiting -- the method would be more powerful and widely applicable were it able to work with just a scene-graph based on the goal scene.
>
> This is a very constructive suggestion. Specifying the final scene as a scene graph would certainly be a generalization. Further, learning more complex scene graph representation (e.g. hierarchical graphs or 3D graphs), or combining multi-modality data (e.g. vision with language) to build more informative scene graphs is definitely a promising future direction as an extension for this work.
>
> - In some parts the paper can be a bit dense with extremely long paragraphs, and it seemed that it could have been edited a bit more to make reading easier.
>
> We have made several edits to make the reading easier.
>
> **Zip File:**
>
> /attachment/ee85838543df8ac96681ddd62a352f75edd08455.zip

---

> > ### Comment · Reviewer_Exub · 2022-08-23
> > **Response to Review Response**
> >
> > Thank you for the addressing my points in the review. I have one follow up question - as to this comment:
> >
> > "In our framework, we use SimGNN for object sequencing, where the network almost always chooses to remove non-target objects first before manipulating target objects since they cause the most difference between the current and the goal scene graph. So the simple hand-coded heuristic approach will achieve similar performance with our approach in most cases. However, in scenarios where the workspace is extremely cluttered and some of the non-target objects are not accessible for grasp while some target objects are available, such approach might fail to complete the task or need more planning steps while the learning-based object sequencing combined with the feature-based action selection will choose to push or move the available target object first."
> >
> > Is that not a significant issue for your submission? If your current experimental setup cannot be used to demonstrate that the SimGNN technique is meaningfully better than this hand-coded heuristic, it weakens the significance of this contribution considerably in my opinion.

---

> > > ### Author Response · Authors · 2022-08-23
> > > **Response to Reviewer Exub**
> > >
> > > Thank you for your response! We really appreciate the follow-up, and we are more than pleased to further clarify on this.
> > >
> > > We do not think it is a significant issue for our submission. Using SimGNN for object sequencing is not only for target/non-target object differentiation. The objective for SimGNN is to prioritize rearranging the vertices (objects) makes the largest difference between the current scene graph and the goal scene graph topology.
> > >
> > > Our observation is when there exists non-target objects in the workspace, SimGNN finds the vertices that are not included in the goal scene graph makes the most topological difference, so it chooses to remove non-target objects before manipulating others. And, when there are only target objects, it picks next object to rearrange based on the spatial relations indicated by the accessibility graph.
> > >
> > > In *Table 3: Scene Graph Comparisons* in section 4.2 in the main paper, we show both cases where there are 10 objects on the table and all 10 objects are targets (all target objects), and there are 20 objects on the table and only 10 of them are targets (non-target objects included). Using SimGNN with accessibility graph shows a significant advantage on both cases.

---

> > > > ### Comment · Reviewer_Exub · 2022-08-27
> > > > **Final Response**
> > > >
> > > > Thank you for the response. It would still appear to me that a simple heuristic to remove non-target objects first would make sense, if SimGNN does indeed learn to always do that (and if that is a requirement for the task anyway). But the fact that Table 3 shows the benefits of SimGNN even when there are no non-target objects makes this point less important, and I could imagine the general technique to be useful in other problem settings where such a simple heuristic may not be a good idea.

---

### Official Review · Reviewer_C9hA · 2022-08-01

**Originality:** Good
**Technical Quality:** Very Good
**Clarity Of Presentation:** Very Good
**Impact:** 3

**Recommendation:**

Weak Accept: I recommend accepting the paper, but will not argue for my recommendation if the majority of other reviewers have a different opinion.

**Summary:**

The paper proposes an image-based learning framework for object rearrangement. In particular, three major challenges are addressed: (1) Objects are closely located with each other. (2) The planner can determine which objects are subject to rearrangement. (3) The planner can resolve a situation when other objects occupy the goal locations of target objects. The proposed method is evaluated on simulations and real-world experiments and shows its promising performance in terms of success rate, accuracy, and efficiency.

**Issues:**

Major comments:
- Contributions compared to existing methods: As discussed above, please explain in the introduction or related work section why previous methods cannot handle clutter and swap. Unless there are clear reasons, the capabilities of existing methods made in Tables 1 and 2 do not look fair. I believe this would enhance the contributions of this work considerably.
- Object sequencing 1: An exponential number of subgraphs can be generated, prohibiting working with a large number of objects in practice. How long does it take to solve 20 objects in the evaluation?
- Object sequencing 2: As pointed out in the limitations section, a straight line path is very naive. When many objects are located closely, motion planning must be used to avoid collisions. In this regard, the high task completion rates in Table 2 are surprising to me. Incorporating motion planning in the learning framework is not straightforward as it requires the notion of configuration space.
- Object sequencing 3: It is unclear how object sequencing handles swaps in Section 3.3. Please explain this. Also, in Figure 1, it seems to clear non-target objects first. Is it always expected to occur?
- POMDP in Section 3.1: It is confusing how solving a POMDP is related to the framework. Is it referring to deep Q-learning, or are they two separate things?
- Section 3.2: The proposed approach does not seem to find a placement right behind an object as that area will be considered as occlusions. If it is incorrect, please explain this.
- In the evaluation, it is not intuitive what makes the proposed method handle clutter. Is it because of feature-based action selection that makes pixel-wise prediction?

Minor comments:
- Do NeRP and IFOR consider the same assumptions and inputs (e.g. both initial and goal images)?
- I was not able to understand what the authors mean by rotating an image in Section 3.1. How do you maintain the resolution of a rotated image before feeding it into FCN?
- In Section 2, "changes in environment observability increase the difficulty of back-tracking and replanning": I don't understand the relationship between changes in environment observability and backtracking and replanning. Please clarify this.
- In Section 3.3, what is "relabeling of a vertex" for?
- In Section 3.3, it is hard to understand how A* can be used to calculate the GED between graphs.
- In Figure 3, images are too dark to see.
- In Figure 1, there are typos. (1-10) -> (1-11) and (11-15) -> (12-15).
- Typo in Line 223: "needs identify" -> "needs to identify".

**Quality Of The Limitations Section:**

Limitations are addressed clearly

**Reviewer Expertise:**

4: The reviewer is confident but not absolutely certain that the evaluation is correct

**Robotics Focus:**

Sufficient demonstration on hardware

**Strengths And Weaknesses:**

Strengths:
- The proposed framework is novel, and it is convincing why this learning approach should work.
- The results from both simulation and experiments look promising.

Weaknesses (details are included in the Summary Of Recommendation and Issues sections):
- Contributions seem less significant as the justification of the fundamental limitations of the previous methods is not well addressed.
- Object sequencing has some critical limitations, which may fail in practice in environments containing many objects.

**Summary Of Recommendation:**

The paper is well-written overall, and the presentation is clear. Although the evaluation results are promising, I am concerned that the contributions may not be significant enough compared to existing methods. The authors state that the previous methods cannot handle the challenges considered in this work as those papers do not treat these challenges. However, that does not mean those methods fail completely with clutter and swap. Since the authors did not evaluate those methods themselves, a more substantial justification is needed by providing insights into the fundamental limitations of those methods. A novel extension to the previous methods is selectivity, which I think is the weak component in the framework, but the same principle also seems to work for them. Moreover, the current setting of the problem having a bin that is not a part of the workspace makes selectivity not a very challenging task.

---

> ### Author Response · Authors · 2022-08-22
> **Response to Official Review of Paper147 by Reviewer C9hA**
>
> **Comment:**
>
> Please note that our response has been split into two parts due to the space constraints. (Part 1/2)
>
> We thank the reviewer for constructive feedback. We are pleased to hear that you found our framework novel and our experimental results promising. You have raised some interesting questions and we would like to discuss them further. Please let us know if any further clarifications are needed.
>
> **Major comments:**
>
> - Contributions compared to existing methods:
>
> This is a great point! In our work, we propose a framework that can solve extremely cluttered tabletop rearrangements. For a highly-cluttered tabletop, direct grasp of an object might not be accessible. A simple example of such cases is shown in Figure 1 in supplementary material where the clutter coefficient is larger than one. When the gripper is not wide enough to grasp more than one object, no grasp is available if the objects are not separated from the clutter first. We introduced push as an additional action primitive in our system for the purpose of separating objects when no direct grasp available in dense clutters. To our best knowledge, no previous object rearrangement systems include push in their action space to benefit grasp success in dense clutter.
>
> Swap is considered a challenging case in object rearrangement tasks. To clarify: we do not claim that no previous work can tackle swap. Methods such as [6], [10], [24] and [25], allocate dedicated buffer locations when swap actions are needed by sampling in free space. In contrast we calculate such locations by directly using the depth information. This prevents our system from placing grasped objects on top of occupied locations (L139-155 in the main paper) which allows pixel-wise sampling in the work space.
>
> - Object sequencing 1:
>
> Algorithm 2 in section 3.3, for N object scenes, only N subgraphs are generated by removing each object vertex in the scene individually. Each inquiry through SimGNN takes ~0.1s, for 20 objects, the object sequencing step takes ~2 seconds.
>
> - Object sequencing 2:
>
> We acknowledge the limitation in our system when motion planning is not incorporated in the learning framework. Our system predicts action location and orientation for robot execution and we use pybullet built-in inverse kinematics (in simulation) and frankx (on physical robot) for motion planning. Incorporating motion planning into the learning process is interesting to explore as future research.
>
> The motion planning failures we are experiencing in physical experiments are mainly from joint limits being reached, mostly from the last joint of the panda robot which can only rotate for a limited amount, in simulated experiments we removed the joint limits which solved some motion planning failures.
>
> - Object sequencing 3:
>
> L139-155, we explained how we were able to deal with swaps. Object sequencing mechanism is only used to determine which object to manipulate next, which does not include swap. Swap is considered in the placing policy where we leverage depth information to avoid occupied pixels to place the currently grasped object.
>
> The non-target objects are almost always moved before manipulating target objects based on the output from SimGNN. Since non-target objects do not exist in the goal scene graph, Algorithm 2 in section 3.3 will almost always choose to remove such vertices in the current scene graph first, which results in non-target objects being almost always being removed first. However, in scenarios where the workspace is extremely cluttered and some of the non-target objects are not accessible for grasp while some target objects are available, the system will choose to push or move the available target object first.
>
> - POMDP in Section 3.1:
>
> The POMDP formulation is only related to feature-based action selection (3.1). Since the observations (raw RGBD images) are not the same as the state (object poses), the problem is a POMDP. However, the observation directly reflects the current state, so we can use deep Q-learning here to solve the POMDP.
>
> - Section 3.2:
>
> That is correct. Our vision-based approach cannot find a placement that is not visible (i.e. occluded) since our placement is based on the visual similarity between the current grasped object and the goal scene. If in the goal image, an object is occluded, our system will not be able to find a placement.
>
> - In the evaluation, it is not intuitive what makes the proposed method handle clutter. Is it because of feature-based action selection that makes pixel-wise prediction?
>
> We introduced push action in our framework (feature-based action selection) to handle clutter. In Figure 3(a), we show that we choose between two actions (push/grasp), if the scene is cluttered, the robot has a lower Q-value for grasp and a higher Q-value for push. Hence in such a case, the robot will execute a push action that separates objects from the clutter, and enables future grasps.
>
> **Zip File:**
>
> /attachment/10fcc094322fcb3c3e935eb814203322560b4465.zip

---

> > ### Comment · Reviewer_C9hA · 2022-08-26
> > **Follow-up question**
> >
> > Thank you for carefully addressing my comments. Most of my questions are clarified, and I have one follow-up question, which I think is critical.
> >
> > One potential weakness of this work would be that a particular scenario of the object rearrangement task is considered, where objects that are irrelevant to a goal can be removed to a bin so that only relevant objects are left in the workspace (i.e. not a cluttered workspace anymore). The existence of the bin seems to simplify the rearrangement task greatly and makes me wonder if the proposed method can be applied to general rearrangement tasks, such as rearranging objects in one cluttered workspace without the existence of the bin. As long as push and pick can be distinguished, motion planning becomes easier in this scenario. That is why a straight line motion was fine (the methods in response to Object sequencing 2 are not really motion planning as they cannot handle obstacle avoidance). Are there any fundamental challenges needed to be addressed for the proposed method, or can the same method handle any general rearrangement tasks? If the latter is the case, I think stating that when introducing the scenario would be helpful.

---

> > > ### Author Response · Authors · 2022-08-27
> > > **Response to Reviewer C9hA**
> > >
> > > Thank you, we’re very happy to hear that most of your questions are clarified.
> > >
> > > Our experiments cover the scenario you describe. In several of our experiments, after removing the irrelevant objects to the side bin, the remaining relevant objects (i.e. target objects) *still form a cluttered arrangement that our system is able to declutter by separating objects (not removing them from the workspace) and complete the task.* For example, in a 15 object experiment, after removing 3 irrelevant objects, the remaining 12 target objects still formed a densely cluttered arrangement. Our system handles this as follows:
> > > 1. if direct grasp available, execute grasp with the highest Q-value, otherwise singulate objects from the clutter for grasp by executing push and then grasp.
> > > 2. if the grasped object’s goal position is not occupied, place it at its goal position (highest cross-correlation), otherwise place it at the nearest empty location (explained L139-155 in the main paper).
> > >
> > > By repeating the 2 steps mentioned above, our system rearranges a cluttered initial arrangement. Though removing irrelevant objects from the workspace does decrease the clutter coefficient of the scene somewhat, it is not the case that binning objects solves the clutter problem. To further check this we have performed additional experiments where *all objects in the initial cluttered scene needed to be rearranged in the final scene (no binning)* and our system works well in those settings. In 43 non-selective 10-object scenarios, with clutter coefficients > 1.0 (1.03 ± 0.02), our system succeeds in 33 trials with 27.73 ± 4.27 planning steps and position error of 1.22 ± 0.17 cm. When introducing the scenario we will be sure to point out that the system is general enough to handle these cases - thank you very much for the suggestion. However, as mentioned in the limitation section, we are cognizant that our system struggles with tight goal arrangements (i.e. where objects have to be placed very near each other in the final configuration).  We are working on addressing this issue in our future work as an extension of the proposed method.

---

> ### Author Response · Authors · 2022-08-22
> **Response to Official Review of Paper147 by Reviewer C9hA**
>
> Please note that our response has been split into two parts due to the space constraints. (Part 2/2)
>
> **Minor comments:**
>
> - Do NeRP and IFOR consider the same assumptions and inputs (e.g. both initial and goal images)?
>
> Yes, that is correct. It is also the reason why we directly compare to their results in Table 2.
>
> - I was not able to understand what the authors mean by rotating an image in Section 3.1. How do you maintain the resolution of a rotated image before feeding it into FCN?
>
> We pad the images while maintaining the same resolution so that after rotation no original image data is lost.
>
> - In Section 2, "changes in environment observability increase the difficulty of back-tracking and replanning": I don't understand the relationship between changes in environment observability and backtracking and replanning. Please clarify this.
>
> This refers to the highly cluttered environment, when a robot takes actions, may reveal more objects or block some objects, which is difficult for classical hierarchical TAMP methods because most planners rely on consistency in instance representations on the high-level.
>
> - In Section 3.3, what is "relabeling of a vertex" for?
>
> Relabeling is referring to “an edit operation” in the beginning of the same sentence.
>
> - In Section 3.3, it is hard to understand how A* can be used to calculate the GED between graphs.
>
> We applied the algorithm introduced as Algorithm 1 in this paper: Riesen, Kaspar, Stefan Fankhauser, and Horst Bunke. "Speeding Up Graph Edit Distance Computation with a Bipartite Heuristic." MLG. 2007., which also is included in our reference [41].
> And used this github repo for implementation: https://github.com/dmlc/dgl, https://github.com/dmlc/dgl/tree/master/examples/pytorch/graph_matching
>
> - We updated the paper pdf to fix the typos and dark figures.

---

### Meta-Review · Area_Chair_S1dU · 2022-08-13

**Recommendation:** Accept (Poster)
**Confidence:** 4

**Metareview:**

The paper proposes a framework for object rearrangement in clutter given a goal image, using graph-based object sequencing, action selection (pushing or grasping), and a placement module. The reviews are mixed, and the current rating of the paper is: weak reject, strong accept, weak accept, weak reject. Reviewers agree that the object sequencing method is interesting, and that the real robot videos are impressive and compelling. However, reviewers also raise several valid questions and concerns including:

* Experiments do not appear to be calibrated: comparisons to prior work should be performed in the same experimental setup for an informative comparison.
* Clarification on novelty claims: why the proposed algorithm is better positioned to handle clutter and swap.

Authors are encouraged to respond to reviewers' comments and questions.

Updates:

Reviewers appreciate the authors' detailed responses, and agree that the presented ideas are interesting, with compelling real robot results on cluttered tabletop rearrangements that involve both translation and rotation. The idea of distinguishing pick and push for rearrangement also adds value to the community. The experiments and ablations in simulation provide helpful quantitative evidence to the efficacy of the approach. There are still a few outstanding concerns on whether the comparisons to prior methods are perfectly calibrated – the authors have clarified that to address this, they have tried to match the experimental setup from prior work as closely as possible: including same data input format, data generation process, and evaluation metrics. Given that the prior methods did not release code for the authors to run on their system, these experiments should suffice (effectively demonstrating the same task but with clear notable improvements).

---

> ### Author Response · Authors · 2022-08-22
> **Response to Meta Review of Paper147 by Area Chair S1dU**
>
> **Comment:**
>
> We thank the meta-reviewers for their insightful comments. We have made some revisions to the paper and supplementary material, and we have replied to each reviewer's comments. We also would like to reply to the meta-reviewer's comments below.
>
> - Experiments do not appear to be calibrated:
>
> We did not reimplement the previous state-of-the-art object rearrangement systems ([24] with translation only and [25] when rotations are considered) because without access to their codebase and other software/hardware configuration details, we found it risky to replicate their systems when there are numerous implementation details (such as physical robot setting, lighting, hyperparameters, training time, random seeds, and hardware) that can largely influence the overall performance of the system. However, we tried to mirror their experimental setting as close as possible in order to provide a calibrated and informative comparison between our system and previous works:
> 1. *Same input format:* images of the current scene and single image of the goal scene.
> 2. *Same data generation process:* randomly place everyday objects (from YCB dataset) on an open tabletop, the number of objects involved are consistent with previous works.
> 3. *Same evaluation matrices:* task completion, planning steps and position error.
>
> Also, we have performed a new experiment with swap only as suggested by reviewer W89x. Our system handles both translation and rotation while NeRP is restricted to translation in object repositioning, which is demonstrated qualitatively in [25]. Despite this handicap, our system produces swap-only results that are comparable to NeRP.
>
> We have added the new results in the supplemental material.
>
> - Clarification on novelty claims: why the proposed algorithm is better positioned to handle clutter and swap.
>
> In our work, we propose a framework that can solve extremely cluttered tabletop rearrangements. For a highly-cluttered tabletop, direct grasp of an object might not be accessible. A simple example of such cases is shown in Figure 1 in supplementary material where the clutter coefficient is larger than one. When the gripper is not wide enough to grasp more than one object, no grasp is available if the objects are not separated from the clutter first. We introduced push as an additional action primitive in our system for the purpose of separating objects when no direct grasp available in dense clutters. To our best knowledge, no previous object rearrangement systems include push in their action space to benefit grasp success in dense clutter.
>
> Swap is considered a challenging case in object rearrangement tasks. To clarify: we do not claim that no previous work can tackle swap. Methods such as [6], [10], [24] and [25], allocate dedicated buffer locations when swap actions are needed by sampling in free space. In contrast we calculate such locations by directly using the depth information. This prevents our system from placing grasped objects on top of occupied locations (L139-155 in the main paper) which allows pixel-wise sampling in the work space.
>
>
> **Zip File:**
>
> /attachment/8a09c64235737123a7ecfc7d015252e80158f491.zip